# Open architecture of archaea MCM and dsDNA complexes resolved using monodispersed streptavidin affinity CryoEM

Jianbing Ma[1,2,3,9], Gangshun Yi [1,4,5,9], Mingda Ye [6], Craig MacGregor-Chatwin[4], Yuewen Sheng [4], Ying Lu [3], Ming Li [3], Qingrong Li [7], Dong Wang [7], Robert J. C. Gilbert [1,5] & Peijun Zhang [1,4,8] ✉

The cryo-electron microscopy (cryoEM) method has enabled high-resolution structure determination of numerous biomolecules and complexes. Nevertheless, cryoEM sample preparation of challenging proteins and complexes, especially those with low abundance or with preferential orientation, remains a major hurdle. We developed an affinity-grid method employing monodispersed single particle streptavidin on a lipid monolayer to enhance particle absorption on the grid surface and alleviate sample exposure to the air-water interface. Using this approach, we successfully enriched the *Thermococcus kodakarensis* mini-chromosome maintenance complex 3 (MCM3) on cryoEM grids through biotinylation and resolved its structure. We further utilized this affinity method to tether the biotin-tagged dsDNA to selectively enrich a stable MCM3-ATP-dsDNA complex for cryoEM structure determination. Intriguingly, both MCM3 apo and dsDNA bound structures exhibit left-handed open spiral conformations, distinct from other reported MCM structures. The large open gate is sufficient to accommodate a dsDNA which could potentially be melted. The value of mspSA affinity method was further demonstrated by mitigating the issue of preferential angular distribution of HIV-1 capsid protein hexamer and RNA polymerase II elongation complex from *Saccharomyces cerevisiae*.

Recently, single particle cryo-electron microscopy (cryoEM) has emerged to become the primary tool for high-resolution structure determination of macromolecules[1]. However, there are challenges in sample preparation, especially for those protein complexes that are not abundantly present in cells and difficult to overexpress, resulting in a very limited amount of material and/or very low attainable concentration, thus seriously affecting their amenability for single particle cryoEM structural analysis[2]. In addition, a well-recognized problem for single particle cryoEM sample preparation is that macromolecular complexes exposed to the air-water interface (AWI) may suffer from denaturation, dissociation, and/or preferential angular distribution[3,4]. It has been reported that 90% of samples would be attracted to the AWI of cryoEM grids using a typical cryoEM sample preparation method[5], demonstrating the significance of the issue.

[1]Division of Structural Biology, Wellcome Centre for Human Genetics, University of Oxford, Oxford, UK. [2]CAS Key Laboratory of Infection and Immunity, National Laboratory of Macromolecules, Institute of Biophysics, Chinese Academy of Sciences, Beijing, China. [3]Beijing National Laboratory for Condensed Matter Physics, Institute of Physics, Chinese Academy of Sciences, Beijing, China. [4]Diamond Light Source, Harwell Science and Innovation Campus, Didcot, UK. [5]Calleva Research Centre for Evolution and Human Sciences, Magdalen College, University of Oxford, Oxford, UK. [6]Centre for Medicines Discovery, Nuffield Department of Medicine, University of Oxford, Oxford, UK. [7]Division of Pharmaceutical Sciences, Skaggs School of Pharmacy and Pharmaceutical Sciences, University of California San Diego, La Jolla, CA, USA. [8]Chinese Academy of Medical Sciences Oxford Institute, University of Oxford, Oxford, UK. [9]These authors contributed equally: Jianbing Ma, Gangshun Yi. ✉e-mail: peijun.zhang@strubi.ox.ac.uk

DNA replication is a fundamental process for cells, during which MCM helicases play essential roles in the initiation and elongation steps[6,7]. MCM helicases are hexameric ring helicases which belong to Superfamily 6 (SF6) of the AAA+ ATPases[7,8]. An MCM monomer is made of two halves, the N-terminal domain (NTD) and the C-terminal domain (CTD). The NTD itself incorporates a helix domain (HD), a Zinc finger domain (ZF), an oligonucleotide binding domain (OBD), and an N-terminal communication loop (NT); while the CDT contains the AAA + catalytic region, a winged helix motif (WH), an exterior hairpin (EXT), a Presensor I β-hairpin (PS1), and a helix 2 insert hairpin (H2I)[6,9–12]. ZF, OBD, NT, PS1, H2I, and WH subdomains were found to be able to bind DNA[6,13–16], while the NT from one monomer also contacts the PS1 of the adjacent monomer[16]. ZF is vital in the formation of double hexamers[17]. The mutation of the ZF could disrupt the double hexamer formation in M. thermoautotrophicum (inclined form) MCM[18] and resulted in lethality or temperature sensitivity in yeast[19,20]. The AAA+ catalytic region of MCM binds and hydrolyzes ATP to drive DNA translocation[6,21]. MCM helicases exist widely in eukaryotes and archaea cells[8]. The WH was reported to regulate the DNA unwinding activity in archaea[12], and the WH of eukaryotic MCM is required during recruitment[10]. In eukaryotic cells, six different homologous MCM proteins (MCM2-7) assemble into a single complex, while in archaeal cells, most species express only one paralog which forms homomeric rings[6]. *Thermococcus kodakarensis* (Tko) serves as a simplified model organism among archaea, which has been valuable to provide insights into the more intricate process of eukaryotic DNA replication[12,21,22]. Tko has three MCM genes (*mcm1-3*), among which *mcm3* is the one essential for DNA replication[23,24]. There have been many efforts to determine the structure of substrate-bound archaea MCM but with little success. The main reason making this complex structure difficult to determine is that the MCM can slide on the dsDNA in the presence of ATP[25,26]. Previous work on other MCM complexes have used very long dsDNA, end-blocked dsDNA, and circular dsDNA to circumvent the problem[25–27]. Recently, Meagher et al. reported structures of truncated and modified archaea *Saccharolobus solfataricus* MCM (SsoMCM) in complex with ssDNA to stabilize the complex[8,28]. However, the challenging structure of MCM in complex with initial substrate dsDNA is still unknown.

To determine the structure of Tko MCM and MCM-dsDNA complexes using single particle cryoEM, we developed affinity grid to resolve the aforementioned issue, as well as alleviate issues with low sample concentrations and AWI effects by enriching and tethering particles on to a substrate surface of the cryoEM grid. A number of affinity interaction pairs have been previously exploited, including Ni-NTA/His-Tag[29–32], antibody/antigen[33,34], streptavidin/biotin[35–39], Spy-Catcher/SpyTag[2]. These interaction pairs were typically prepared on carbon films[2,31,33,34], graphene sheets[40–43] or lipid monolayers[30,35–39], and their successful applications have been reported[44–46]. The ultra-high affinity pair of streptavidin/biotin was utilized to form a streptavidin 2D crystalline lattice on a lipid monolayer, to which the target protein complexes could then attach[44–47]. Although the background of a streptavidin 2D crystalline lattice can be minimized by applying an appropriate Fourier filter[37,38], there were other drawbacks, such as fixed orientation and rigid distances. Here we generated streptavidin affinity grids with monodispersed single particle streptavidin (mspSA) on lipid monolayers, which allowed for high-resolution structure determination of challenging samples.

In this work, we developed mspSA affinity grids and achieved 10-50 times enrichment and determined the structures of biotinylated MCM and MCM in complex with biotin-tagged dsDNA in the presence of ATP. We found both hexameric structures are left-handed spirals in a rare open conformation. The open gate of MCM is sufficiently large for the dsDNA to be loaded, and potentially in a partial melted conformation, primed for DNA replication initiation[9,48–50]. We further demonstrated the utility of the mspSA affinity method beyond the sample enrichment, by mitigating the issue of preferred orientation in two distinct samples: HIV-1 capsid protein (CA) hexamers and RNA polymerase II elongation complex from *Saccharomyces cerevisiae* (ScPol II EC), allowing an improvement of both structures.

## Results

### Monodispersed single particle streptavidin (mspSA) affinity grid enriches archaea MCM complexes

Owing to the high affinity ($K_d \approx 10^{-15}$ M) between streptavidin (SA) and biotin, we selected this pairwise interaction to prepare affinity grids for cryoEM. Inspired by the earlier work[51], we constructed the affinity grids using biotinylated lipid monolayer for SA (from *Streptomyces avidinii*) attachment (Fig. 1a, Supplementary Fig. 1). We have tested a wide range of conditions, including SA concentrations, ratio of biotin-lipid, salt, pH, and obtained a phase diagram for the SA conformation on the lipid monolayer (Fig. 1b). At low SA concentration or low biotin/lipid ratio, SA forms monodispersed single particles (Fig. 1c, Supplementary Fig. 2), while at high SA concentration and high biotin/lipid ratio it forms an extended 2D crystalline lattice which has been previously reported (Fig. 1d, Supplementary Fig. 2)[38]. At 0.08 mg/ml SA concentration and 15% biotinylated lipid, the formation of mspSA is robust and consistent, whereas the formation of 2D crystalline SA occurs consistently at 0.15 mg/ml SA concentration and above 25% biotinylated lipid. The mspSA affinity grid has advantages over the 2D crystalline SA affinity grid[37,38], overcoming restraints on the particle orientation due to the rigid 2D SA lattice. We thus exploited the mspSA affinity grids for cryoEM sample preparation of Tko MCM helicase and with its dsDNA substrate.

To enrich the MCM helicase on cryoEM grids, MCM was randomly biotinylated at its surface lysine residues (Fig. 2a and Supplementary Fig. 3). This enabled us to obtain suitable cryoEM grids at low MCM concentration (0.05 mg/ml). The average number of biotinylated MCM particles on the affinity grid was 316.2 per micrograph, compared to 7.5 particles per micrograph without biotinylation (Figs. 1e and 2a, b). This result shows a great enrichment of MCM (42 times) using mspSA affinity grids, which enabled cryoEM structural analysis of MCM at a low concentration of 0.08 mg/ml (Fig. 2).

For the MCM helicase loaded on dsDNA, we used a different strategy. We tagged both ends of 200 bp dsDNA with biotin to anchor the dsDNA onto the mspSA grids thus preventing loaded MCM from sliding off the dsDNA in the presence of ATP (Fig. 3a). At a low MCM concentration of MCM (0.07 mg/ml), the affinity grid gave rise to an average of 257.5 particles per micrograph (Figs. 1e and 3b), compared to the average of 22.5 particles per micrograph from the control sample with un-tagged dsDNA substrate (Fig. 1e and 3b), a significant improvement (11 times) of the MCM-ATP-dsDNA complex.

### Structure determination of MCM and MCM-ATP-dsDNA complexes

The 2D classes of MCM without dsDNA (MCM-apo) showed clear MCM ring features without apparent streptavidin (60 kDa) density, suggesting that the mspSA is randomly associated with MCM-apo and has been averaged out (Fig. 2c). The MCM-apo forms double hexameric rings, but flexible relative to each other (Fig. 2c and Supplementary Fig. 4a). We obtained a cryoEM map of MCM-apo hexamer at an overall resolution of 3.26 Å (Fig. 2 and Supplementary Fig. 4). As indicated by the local resolution, the MCM-apo is highly flexible, especially at the two peripheral MCM subunits (Supplementary Fig. 4d). The central four MCM monomers are better resolved (2.5-3.0 Å resolution). Similarly, the cryoEM map of MCM-ATP-dsDNA complex was reconstructed at an overall resolution of 3.57 Å (Fig. 3 and Supplementary Fig. 5), while the local resolution of the central MCM monomers is 3.0-3.5 Å, sufficient to distinguish ATP from ADP (Supplementary Fig. 6a).

The conformation of the ATP pocket located at interfaces of neighboring subunits (Fig. 3e, bottom panel) often correlates with DNA and ATP binding and hydrolysis. In the MCM-ATP-dsDNA

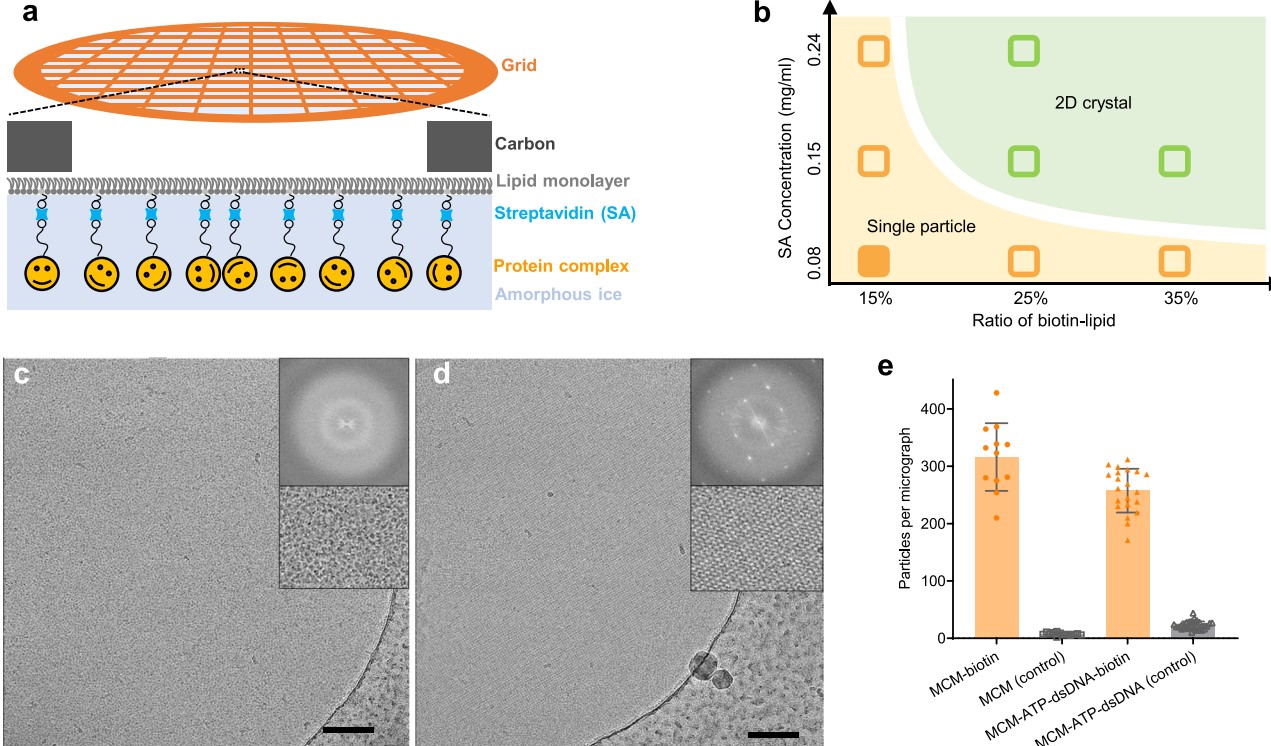

**Fig. 1 | Affinity grid of single particle distributed streptavidin (SA). a** A schematic diagram for generating the SA affinity grid. **b** The phase diagram of SA organization on EM affinity grids. **c, d** Typical micrographs of monodispersed single particle SA (mspSA) (**b**) and 2D lattice SA (**c**). The scale bars are 100 nm. **e** Comparison of the number of MCM complexes in each micrograph without (gray, $n = 13$) and with (gold, $n = 12$) biotinylating of MCM (left), and MCM-ATP-dsDNA complexes without (gray, $n = 31$) and with biotin-tags on dsDNA (gold, $n = 21$) (right). The error bars are standard errors. Source data are provided as a Source Data file.

structure, the interfaces of A-B, B-C, C-D, D-E, and E-F are occupied by two ATP (C-D, D-E) and three ADP (A-B, B-C, E-F) molecules (Supplementary Fig. 6a). Like other AAA+ helicases, the AAA+ catalytic region of Tko MCM contains the conserved motifs including the Walker-A and -B motifs, Sensor 1 and 2, and an arginine-finger[21]. In comparison with MCM-apo, the ATP pocket of MCM-ATP-dsDNA showed several differences. First, at the interface of A-B, a loop of A subunit in the ATP pocket (R329-334, squared in Supplementary Fig. 6b) moved about 2 Å. Second, at the subunit interface B-C, both the loop of B (R329-334) and the sensor-2 (R555) of C subunit shifted. Third, at the D-E interface, the changes are much more substantial at the positions of two alpha helixes where the sensor-2 (R555) and Arginine finger (R462) are located (Supplementary Fig. 6b).

Compared to the conventional cryoEM samples, mspSA affinity grids contain additional background (lipid and SA) that could affect the alignment of the particles, especially those with lower molecular weights. To evaluate this effect, we utilized the particle subtraction algorithm in CryoSPARC (v3.3.2)[52] to generate subcomplexes of different sizes, tetramer (306 kDa), trimer (230 kDa), and dimer (153 kDa) MCM datasets (Supplementary Fig. 7) from hexamer MCM, and carried out alignment of these subcomplexes with routine SPA analysis. CryoEM maps of these subcomplexes were obtained for the tetramer at 3.37 Å, trimer at 3.50 Å, and dimer at 5.90 Å, respectively. These results indicate that the mspSA affinity grids are also applicable to smaller particles.

### Both MCM-apo and MCM-ATP-dsDNA adopt left-handed open spiral conformations

The MCM-apo structure shows six MCM monomers, interestingly, arranged as a left-handed open spiral (Fig. 2d-e). We labeled the six monomers of MCM from top to bottom as A, B, C, D, E, and F, colored

from red to blue as illustrated (Fig. 2e). The spiral conformation, central channel diameter, and open gate size are the key parameters to distinguish this MCM structure from other previously reported MCM structures, as summarized in Table 1 (more details in Supplementary Table 1). First, the MCM-apo is a left-handed open spiral with a substantial rise per turn of 46 Å, whereas other archaeal MCM helicases (PDB: 4R7Y, 8EAL and 8EAM)[8,53] are either closed or right-handed open with a small rise (10-15 Å, measured at the H2I and PS1 loops). A left-handed open spiral was observed from human MCM, but with a smaller rise (22-34 Å, PDB: 7W68) (Table 1). Second, the central channel diameter at the CTD of MCM-apo is ~50 Å which is much larger than other reported archaeal and eukaryotic MCMs (20-25 Å for closed hexamer to 33-35 Å for open hexamer) (PDB: 7W68, 5XF8) (Table 1, Supplementary Table 1). Third, the opening gate size of MCM-apo is 20 Å, larger than previously reported semi-open conformations of MCM from archaeal (PDB: 8EAM) and eukaryotic (4.4 Å, PDB: 7W68) species (Supplementary Fig. 8). The structure of MCM-ATP-dsDNA also shows a left-handed open spiral with a rise of 46 Å (Fig. 3d-e). The central channel diameter at the CTD of MCM-ATP-dsDNA measures 45 Å and gate size 15 Å, slightly smaller than MCM-apo. Both MCM-apo and MCM-ATP-dsDNA form head-to-head double hexamer but are very flexible at the hexamer-hexamer interface (Figs. 2c and 3c), in stark contrast to previously reported rigid double hexamers of eukaryotic MCM in a closed conformation. MCM-ATP-dsDNA appears to show better density of the stacked hexamer compared to MCM-apo (Figs. 2c and 3c), suggesting that dsDNA loading constrains the double hexamer configuration.

The MCM-apo and MCM-ATP-dsDNA comprise an N-terminal domain (NTD), a C-terminal domain (CTD), and the same subdomains as found in other MCM helicases. There is little deviation among the six subunits for both MCM-apo and MCM-ATP-dsDNA (Figs. 2g and 3f,

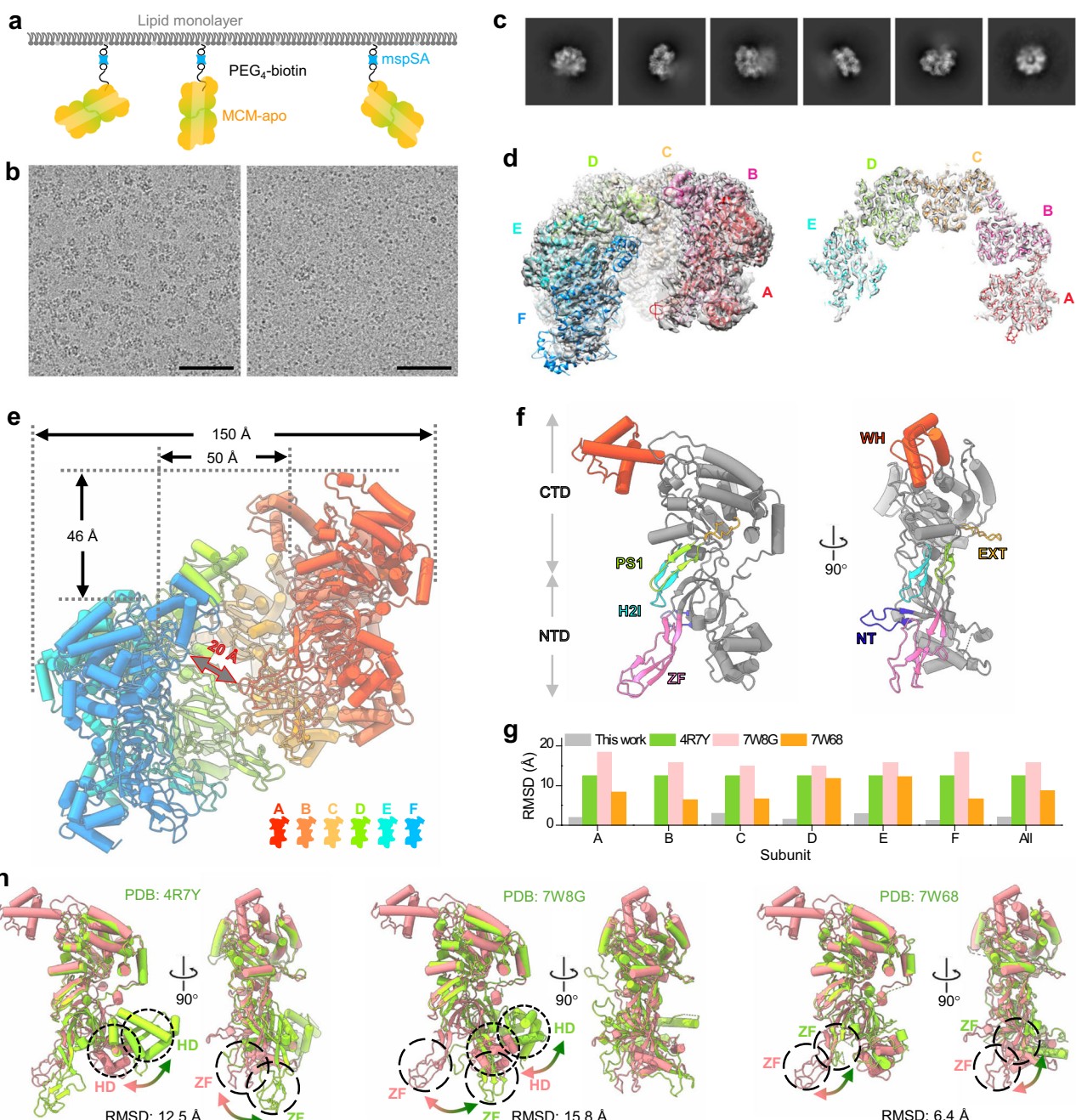

**Fig. 2 | CryoEM structure of MCM-apo captured by mspSA affinity grids. a** The design of mspSA affinity-capture of biotinylated MCM-apo. **b** Representative micrographs of MCM-apo on mspSA affinity grids at 0.05 mg/ml concentration with (left) and without (right) biotinylation. The SA used is 0.02 mg/ml. Scale bars, 100 nm. The experiment was repeated two times independently with similar results. The number of image is 13 (without biotinylation) and 12 (with biotinylation). **c** 2D classes of MCM-apo show very flexible double MCM hexamers, one clear and the other blurry. **d** A side view (left panel) and a middle slice (right panel) of the cryoEM map of the MCM-apo hexamer at 3.26 Å resolution, overlapped with the atomic model of each subunit, colored separately. The model of the best-resolved subunit was first built and refined, which was subsequently used to refine all other subunits. The peripheral subunit (F, in blue) is flexible and less resolved. **e** Structure of MCM-apo hexamer with each subunit colored according to the color

key. The size of hexamer and spiral rise were measured. The size of center channel at CTD was measured between the main chains of two amino acids at the same Z position at CTD. The gate size was measured between the main chains of the nearest two amino acids from subunit A (ZF) and F(CTD). **f** Structure of a single MCM-apo subunit B which is best-resolved. Structural motifs of ZF, WH, NT, H2I, PS1, and EXT are highlighted with color. **g** Root Mean Square Deviation (RMSD) between the B subunit of MCM-apo and the B subunit of other published MCM structures (colored accordingly). The gray bar indicates the averaged RMSD between the B subunit and other subunit of MCM-apo. The structures were aligned on the CTD. Source data are provided as a Source Data file. **h** The B subunit of MCM-apo (pink) superimposed with the B subunit of other published MCM structures (green) aligned on CTD. Large shifts occur at the HD and ZF subdomains (circled), as indicated by arrows.

gray bar), and the subunits are also very similar between MCM-apo and MCM-ATP-dsDNA (Supplementary Fig. 8c). The main difference between MCM-apo and MCM-ATP-dsDNA lies in the appearance of the WH motif: in MCM-apo, the WH motif shows in every other MCM

monomer, i.e., B, D, and F. While in MCM-ATP-dsDNA, the WH motif shows from the second MCM to the last monomer, i.e., B, C, D, E, and F (Supplementary Fig. 8d). This variable degree of ordering in the WH motif is notable and very interesting.

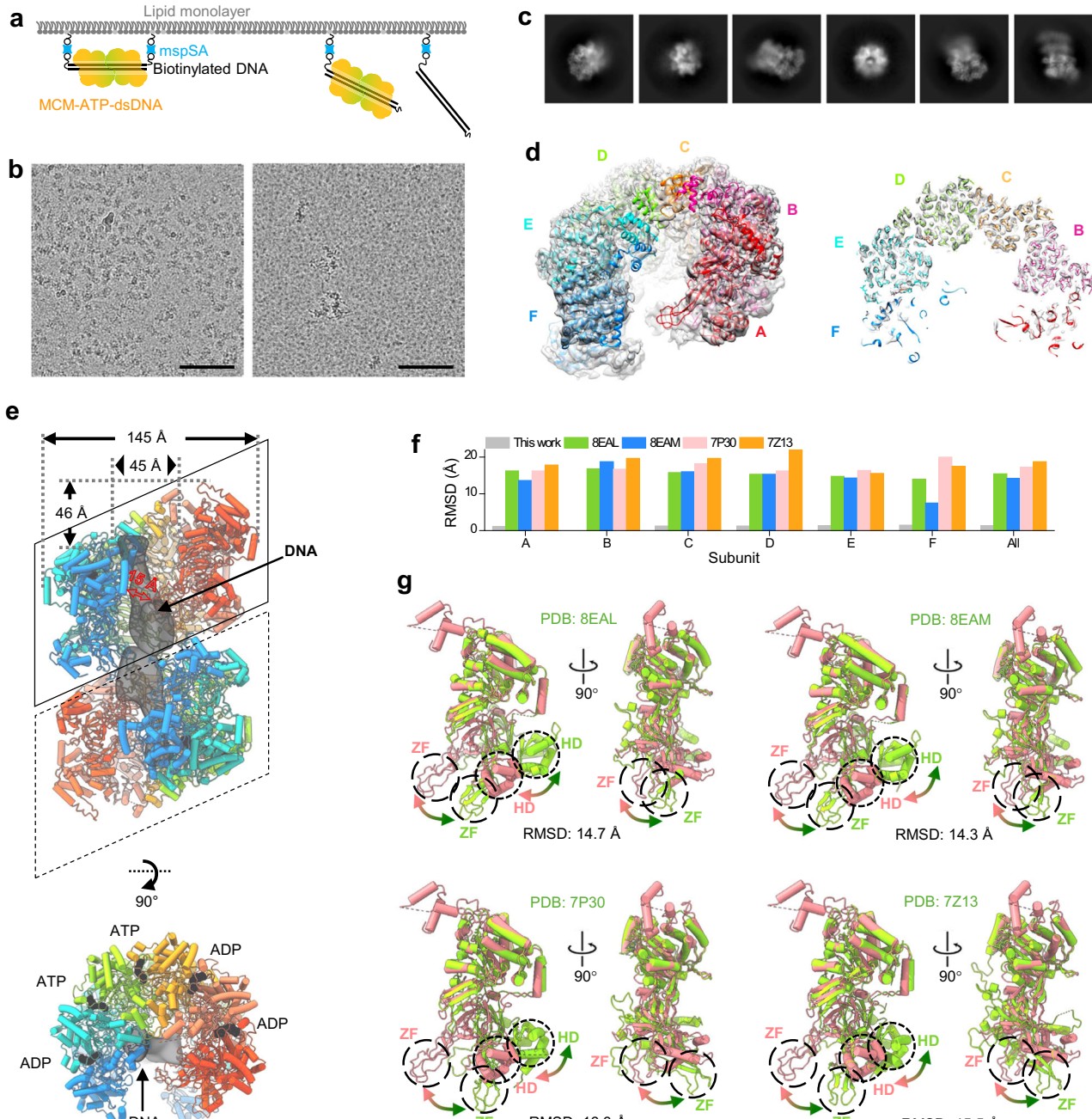

**Fig. 3 | CryoEM structure of MCM-ATP-dsDNA captured by mspSA affinity grids.**
**a** The design of mspSA affinity-capture of biotin-tagged dsDNA in complex with MCM and ATP. **b** Representative micrographs of MCM-ATP-dsDNA on mspSA affinity grids at 0.07 mg/ml concentration with (left) and without (right) biotin-tag on both ends of dsDNA. The SA used is 0.04 mg/ml. Scale bars, 100 nm. The experiment was repeated two times independently with similar results. The number of image is 31 (without biotin-tag) and 21 (with biotin-tag). **c** 2D classes of MCM-ATP-dsDNA show flexible double MCM hexamer. **d** A side view (left panel) and a middle slice (right panel) of the cryoEM map of the MCM-ATP-dsDNA hexamer at 3.57 Å resolution overlapped with the atomic models of subunits (colored). The model of the best-resolved subunit was first built and refined, which was subsequently used to refine all other subunits. The peripheral subunits (both the red and blue subunits) are less well resolved. **e** Structure of MCM-ATP-dsDNA double hexamer, where the bottom MCM was docked using the model from the top MCM. The nucleotide (ADP or ATP) bound between two subunits is clearly resolved. The density of DNA located in the central channel of double MCM hexamers is low passed to 16 Å. The distances were measured with the same method as described in Fig. 2e. **f** RMSD values between the B subunit of MCM-ATP-dsDNA and the B subunit of other published MCM-dsDNA structures (colored accordingly). The gray bar indicates the averaged RMSD between the B subunit and other subunit of MCM-ATP-dsDNA. Source data are provided as a Source Data file. **g** The B subunit of MCM-ATP-dsDNA (pink) superimposed with the B subunit of other published MCM structures (green) aligned on CTD. Large shifts occur at the HD and ZF subdomains (circled), as indicated by arrows.

We further compared MCM-apo subunits with MCM subunits from previous archaeal (PDB: 4R7Y) and eukaryotic (PDB: 7W68 and 7W8G) studies (Fig. 2g, h). MCM structures determined in close conformations (4R7Y and 7W8G) show the largest deviation from our structure (RMSD of 12.5 and 15.8 Å), with HD and ZF subdomains significantly displaced (20-30 Å) (Fig. 2h). The MCM structure in its open conformation (PDB: 7W68) shows a smaller deviation compared to ours (RMSD of 6.4 Å) (Fig. 2h), with no significant displaced subdomain. We also compared MCM-ATP-dsDNA complex with other MCM-DNA complexes from archaeal (PDB: 8EAL and 8EAM, ssDNA)

**Table 1 | Comparison of structure features from various MCM**

| Source | Complex | MCM layer | Resolution (Å) | PDB | Closed/open | Spiral of hexamer | Spiral of H2I and PS1 | Center channel (CTD) | Open gate | Nucleotide | DNA |
|---|---|---|---|---|---|---|---|---|---|---|---|
| Archaea | *Pyrococcus furiosus* NTD | Single hexamer | 2.65 | 4POF | Closed | – | – | 25 Å | – | – | – |
| | *M. Thermoautotrophicum* NTD | Double hexamer | 3.0 | 1LTL | | | | | | | |
| | *Sulfolobus solfataricus* NTD *Pyrococcus furiosus* CTD | Single hexamer | 2.7 | 4R7Y | | | – | | | | ADP: 6 | |
| | *Sulfolobus solfataricus* | | 2.34 | 8EAL | Semi-open | | Right-handed 10 Å | | | ADP-BeF3: 5 | ssDNA |
| | | | 2.59 | 8EAM | Semi-open | | Right-handed 15 Å | | | ADP-BeF: 4 | |
| | ***Thermococcus kodakarensis*** | **Double hexamer** | **3.26** | 8X7T | **Open** | **Left-handed 46 Å** | | **50 Å** | **20 Å** | **–** | **–** |
| | | | **3.57** | 8X7U | | | | **45 Å** | **15 Å** | **ATP: 2 ADP: 3** | **dsDNA (maybe melted)** |
| Eukaryon | Yeast | Cdt1-MCM | Single hexamer | 7.1 | 5XF8 | Open | Left-handed 26-34 Å | Left-handed 34 Å | 35 Å | 6 Å | Apo | – |
| | Human | MCM | Single hexamer | 4.4 | 7W68 | Open | Left-handed 22-34 Å | Left-handed 22 Å | 33 Å | 4 Å | Apo | – |
| | Yeast | | Double hexamer | 3.0 | 7P30 | Closed | – | | 25 Å | – | ATP: 1 ADP: 4 | dsDNA |
| | | | 2.52 | 7W8G | | | | | | | ATPγS: 5 ADP: 1 | – |
| | Human | | 2.59 | 7W1Y | | | | | | | ATP: 4 ADP: 2 | dsDNA |
| | Yeast | Cdc45-MCM-GINS-Pol ε | 3.4 | 7Z13 | | | | | | | ATP: 4 ADP: 2 | dsDNA (melted) |

Representative MCM structures with highest resolution for each conformation, replication initiation stage, or species are shown. Data in bold are from this study. The distances were measured at the main chains of the structures. A more comprehensive comparison with additional MCM structures is shown in Supplementary Table 1.

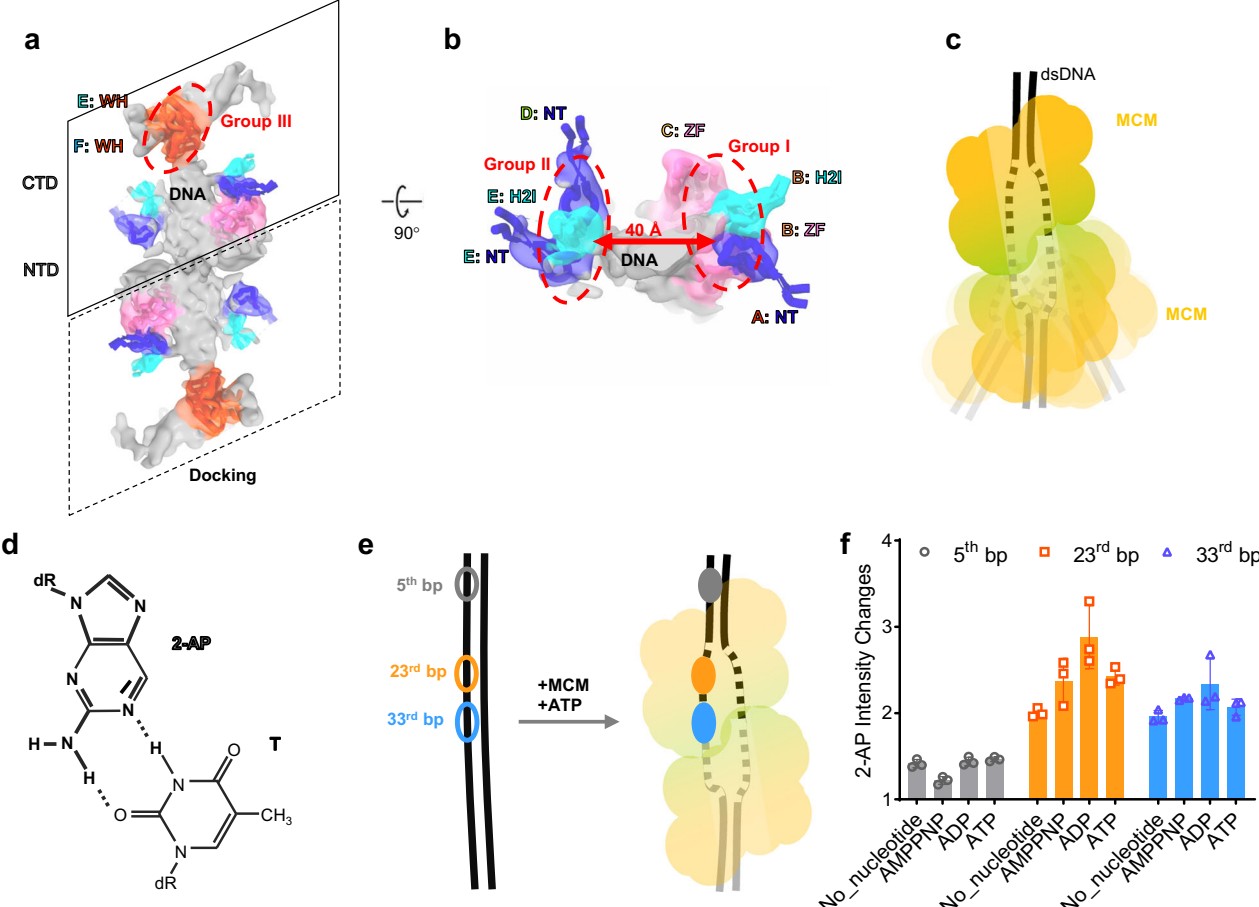

**Fig. 4 | Interactions between dsDNA and double MCM hexamers in the MCM-ATP-dsDNA complex. a, b** The interacting amino acids are clustered into three groups based on their positions. Groups I and II are located at the opposite side of the dsDNA, 40 Å apart; Group III is located at the WH motifs. Only the densities of dsDNA (light gray) and MCM-dsDNA interfacing loops (colored as in Fig. 2f) are shown for clarity. The density was low-passed to 8 Å. **c** A schematic model of the MCM-ATP-dsDNA double hexamers, where the dsDNA is shown in the melted state and the bottom MCM-hexamer is highly dynamic. **d** Skeleton symbol of the 2-aminopurine (2-AP):T base pair. **e** The locations of (2-AP):T base pair on the 66 bp dsDNA template, at the 5th (gray), 23rd (orange), and 33rd (blue) bp, respectively. **f** The intensity changes of 2-AP fluorescence at the three locations and in the presence of different nucleotides. The error bars are standard deviation ($n = 3$). Source data are provided as a Source Data file.

and eukaryotic (PDB: 7P30 and 7Z13, dsDNA) species. The open conformation of MCM-ATP-DNA is distinct from these semi-open conformations (8EAL and 8EAM) or closed conformations (7P30 and 7Z13) (Fig. 3f, g). This is mainly due to the large shift of ZF and HD subdomains (20−30 Å) (Fig. 3g). Moreover, the analysis of the overall shape of the MCM subunit between the open, semi-open and closed states (Supplementary Fig. 9) gave rise to the differences in the central channel size and the handedness of the spiral (Supplementary Fig. 9c).

## MCM-DNA interactions

The observed differences between MCM-apo and MCM-ATP-dsDNA are likely important to DNA binding and potentially dsDNA melting. In the MCM-ATP-dsDNA map, we identified the density of DNA in the central channel of MCM (Fig. 3e, dark gray). This region was not well resolved, but the interaction interfaces between DNA and MCM can be delineated (Fig. 4). We found that MCM makes multiple contacts with the DNA, specifically through subdomain NT from MCM subunit A, ZF and H2I from subunit B, ZF from C, NT from D, NT, H2I, and WH from E, WH from F. These interacting subdomains can be clustered into three groups, groups I-III based on their positions (Fig. 4a, b). Group I involve A (NT), B (ZF and H2I), and C (ZF), group II involve D (NT) and E (NT, H2I), and group III involve E (WH) and F (WH). Only some of H2I, NT, and ZF in the hexamer make contact with the DNA, consistent with previous reports[14,15,54]. This may be due to the larger center channel and

the mismatch handedness between MCM and B-type dsDNA. Intriguingly, the distance between the interaction groups 1 and 2 are about twice as wide as expected from dsDNA, suggesting it might be partially melted (Fig. 4b). Interestingly, the density corresponding to the DNA is much broader than expected (Fig. 4a-b, Supplementary movie 1), further supporting this notion, although DNA flexibility could also contribute to the broadening appearance.

To further verify the dsDNA conformation, we used 2-aminopurine (2-AP) to monitor the base pair melting with its fluorescence (Fig. 4d-f)[55–57]. We placed 2-AP at three positions on the 66 bp dsDNA: 1) near the dsDNA end (5th bp), 2) at the 23rd bp position, and 3) at the 33rd bp position, both close to the center of the complex (Fig. 4e). The 2-AP fluorescence enhanced 2−3 folds at the second and third positions compared to the first position independent of nucleotides (Fig. 4f, Supplementary Fig. 10), supporting the notion of dsDNA local melting.

## The mspSA affinity method effectively resolves the issue of preferential angular distribution

The affinity method prevents particles from contacting the air-water interface (AWI), thus potentially mitigating issues associated with AWI, such as preferred orientation and denaturation. We applied mspSA affinity method to two problematic samples that suffered from severe preferred orientation problem: HIV-1 CA hexamer[58] and ScPol II EC[59].

The HIV-1 CA protein is a critical structural component of the virion and facilitates essential life cycle processes through interactions with host cell factors[58,60–62]. Previous studies of the cross-linked CA Di-hexamer (CA A14C/E45C/W184A/M185A) have reported a significant issue with angular distribution[58]. The biotinylated CA Di-hexamers were subsequently attached to the mspSA affinity grid (Fig. 5a) or prepared on a holey carbon grid as a control. During data processing, numerous particles exhibited evenly distributed 2D projection views when prepared using the mspSA affinity grid (top panel in Fig. 5b). However, the majority of the particles on the holy carbon grid exhibited a top view of the CA Di-hexamer (see bottom panel in Fig. 5b). In both samples, particles displayed stacked CA Di-hexamers (Fig. 5b and Supplementary Fig. 11). A 3.14 Å structure of the stacked CA Di-hexamer was successfully reconstructed for the sample on the mspSA affinity grid (Fig. 5c, d and Supplementary Fig. 11). However, due to the pronounced angular distribution problem, we were unable to reconstruct the structure of CA on the holy carbon grid.

RNA polymerase II is the enzyme responsible for synthesizing all messenger RNAs in eukaryotic cells and constitutes the core component of the transcription machinery[63]. The ScPol II EC is a complex which is composed of 12 subunits (Rpb1 to Rpb12) and 3 strands of DNA/RNA[59] (Fig. 5e). It has been reported to have a significant angular distribution problem[59]. We added biotin tags to 5′-end of both template and non-template DNA strands for the attachment and enrichment of the complex on mspSA affinity grids (Fig. 5e). This preparation method resulted in particles exhibiting evenly distributed 2D projection views (top panel in Fig. 5f). In contrast, most particles on the holey carbon grid tended to adopt the same view, consistent with previous studies (bottom panel in Fig. 5f). A 2.98 Å structure was determined for the sample on the mspSA affinity grid (Fig. 5g, h and Supplementary Fig. 12). However, due to the severe angular distribution problem, we were unable to obtain a 3D reconstruction for the sample on the holey carbon grid with the same number of particles.

## Discussion

One of the main limitations in single particle cryoEM structural determination is sample preparation. It is still challenging to prepare suitable cryoEM grids for low abundance/low concentration samples[2]. Additionally, the AWI is a well-recognized problem affecting numerous samples because it can lead to preferential particle orientation, denaturation or disassembly of protein complexes[3,4]. Affinity grids could alleviate both issues by enriching the particles on to the grid and keeping them away from the AWI. The mspSA grid is robust and simple, easy to adapt. The method can tolerate a wider range of conditions, including SA concentration, buffer condition, and better coverage on the grid. The SA density is usually averaged out in the image processing and reconstruction. A future improvement can be made to reduce the SA background by using engineered SA monomer (13 KDa) or dimer (26 KDa)[64,65]. It is also plausible to design ultra-high affinity interaction pairs with even smaller fragments using AI[66].

To understand the mechanism of DNA replication initiation, many MCM structures from eukaryotic species have been reported and models for their activity proposed[14,15,26,27,67–74]. Essentially the DNA replication initiation begins with a single MCM-Cdt1 loaded onto dsDNA by ORC-Cdc6 in a closed conformation[71]. The replication starts when a head-to-head MCM-Cdt1 double hexamer is formed to melt the dsDNA[15,48]. In yeast, only when double Cdc45-MCM-GINS is formed by recruiting additional factors Cdc45, GINS, and Pol ε, does dsDNA melt[14]. Several structures of archaea MCM from *M. thermoautotrophicum*, *Sulfolobus solfataricus*, and *Pyrococcus furiosus* have been reported[8,28,53,75], in closed or semi-open conformations. By using mspSA affinity-grids, we were able to determine cryoEM structures of both Tko MCM and MCM-ATP-dsDNA complexes in open conformations, thus filling a critical gap in our knowledge regarding DNA replication.

Comparing open MCM from yeast and archaea, one intriguing difference lies in the size of the gate in each case. While Tko MCM-apo constitutes a wide-open gate (20 Å), large enough to fit dsDNA, the open MCM-Cdt1-apo from human and yeast has a very narrow gate (4–6 Å) (Table 1)[70,76]. Another exciting feature in MCM-ATP-dsDNA is a wide central channel which would accommodate the separation of the two strands of DNA. The structure of MCM-ATP-dsDNA complex is distinct from other MCM-ATP-dsDNA structures in closed conformations containing either un-melted[26,27,74] or melted DNA[6,14,15]. From the density map of dsDNA and the large size of the open channel, we speculate that Kto MCM-ATP-dsDNA may contain melted or partially melted DNA, supported by the 2-AP fluorescence measurements (Fig. 4).

Taken together, we developed mspSA affinity method which enabled the structure determination of archaea MCM and in complex with dsDNA, HIV-1 CA Di-hexamer and ScPol II EC, while providing essential insights into DNA replication initiation. The findings underscore the efficacy of the mspSA affinity grid as a robust solution to the complex problems of angle distribution and low sample concentration. The mspSA affinity grids will thus, we hope, be of great interest to a broader research community.

## Methods

### Protein expression and purification

The *Thermococcus kodakarensis mcm3* gene originated as a donation from the lab of Xipeng Liu, located at Shanghai Jiao Tong University in Shanghai, China. The gene was cloned into the pHGT-bio vector (gift from the Centre for Medicines Discovery, University of Oxford) and expressed as an N-terminal His$_6$-GST fusion protein. This protein was expressed in *E. coli* KRX stain by growing the bacteria in TB medium at 37 °C for 12 hours until $OD_{600} > 1$, followed by addition of 0.1% Rhamnose and culturing for another 20 hours at 22 °C. It was then purified using Ni$^{2+}$-affinity chromatography. Following this initial process, the remaining contaminants were effectively removed by passing through a Q HP column (GE) and subsequently eluted with a linear gradient spanning from 0.1 M to 1 M NaCl. The eluted fractions were pooled together and concentrated for size exclusion chromatography (Superdex 200 Increase 10/300 (GE)) with buffer containing 20 mM HEPES (pH 7.8), 100 mM NaCl, 2 mM DTT in a 4 °C cold room. Protein fractions with purity >95% were pooled together and concentrated and stored at -80 °C.

The disulfide designed mutant CA protein (A14C/E45C/W184A/M185A) was clone into pET21a vector[77]. This protein was expressed in *E. coli* Rosetta stain by growing the bacteria in LB medium at 37 °C until $OD_{600} > 0.6$, followed by addition of 0.1 M IPTG and culturing for 20 hours at 22 °C. CA protein was purified by passing through a Q HP colume (GE) and subsequently eluted with a linear gradient spanning from 0 M to 1.5 M NaCl. The fractions containing CA were pooled together followed by adjusting the pH to 5.8 with diluted acetic acid and centrifuged at 48,384×$g$ for 1 h at 4 °C. The supernatant was passed through S HP column (GE) and subsequently eluted with a linear gradient spanning from 0 M to 1 M NaCl. The fractions containing CA were pooled together and concentrated before loading onto a size-exclusion column (HiLoad 16/600 200 prep grade) equilibrated in buffer containing 20 mM HEPES (pH 8.0), 100 mM NaCl at 4 °C in a cold room. Pure CA was cross-linked by the addition of 30 mM CuCl$_2$ aiming to form CA Di-Hexamer at room temperature for 30 min followed by concentration before loading onto a size-exclusion column (Superdex 200 Increase 10/300 (GE)) equilibrated in buffer containing 20 mM HEPES (pH 8.0), 100 mM NaCl at 4 °C in a cold room. Early fractions containing Di-hexamer CA were stored in -80 °C freezer.

The 12-subunits RNA polymerase II (Pol II) from *S. cerevisiae* was purified according to previously established protocols[78,79]. In summary, Pol II, tagged with a recombinant protein A on the Rpb3 subunit, was first subjected to affinity chromatography using an IgG column (Cytiva). This

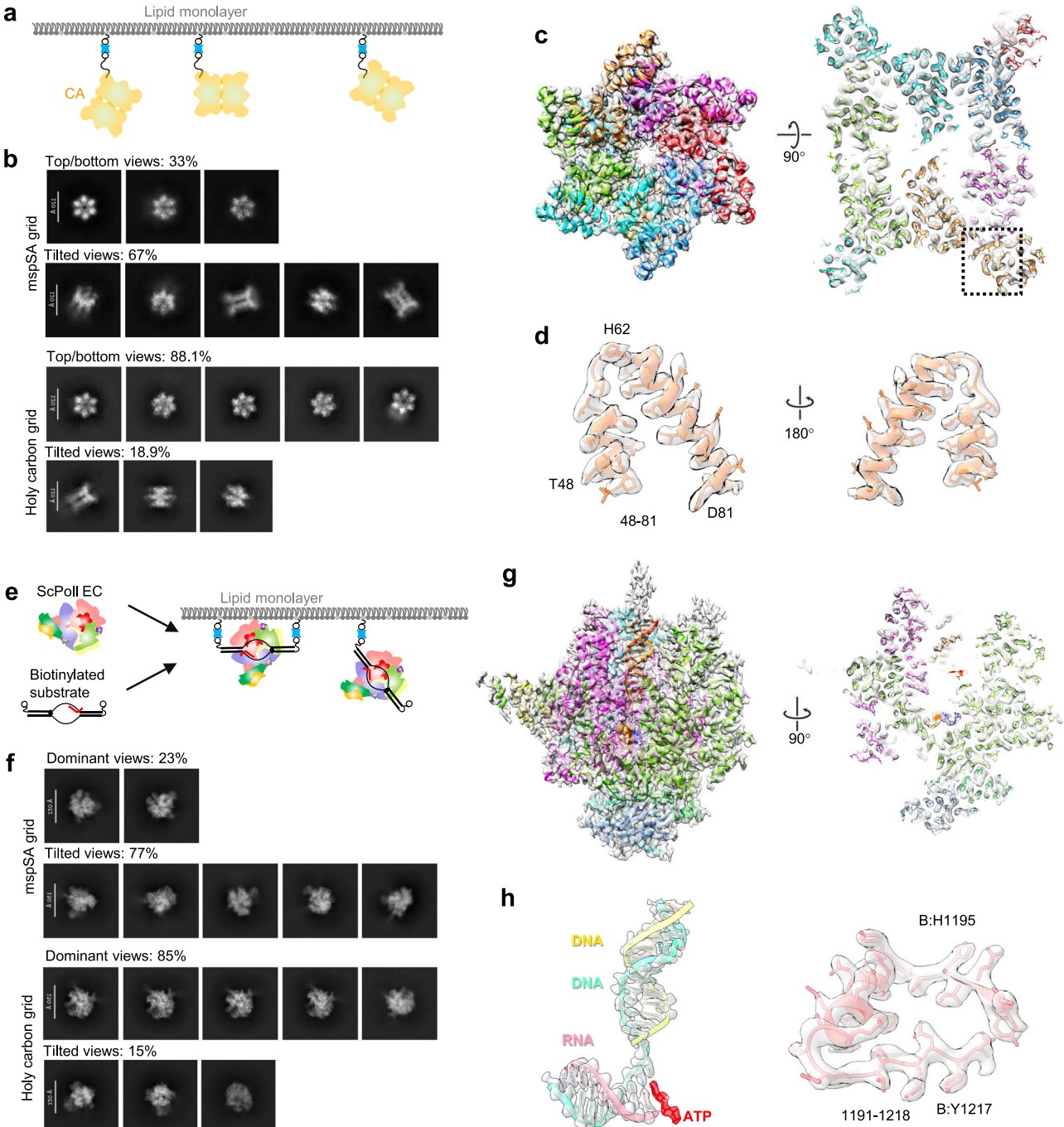

**Fig. 5 | CryoEM structures of HIV-1 CA Di-hexamer and scPol II EC complex.**
**a** The design of mspSA affinity-capture of biotinylated HIV CA Di-hexamer.
**b** Representative 2D classes of HIV-1 CA Di-hexamer generated from 1000 micrographs of mspSA affinity grid and holy carbon grid respectively applied same parameters for blob particle picking. 2D classes are separated into top/bottom- and tilted-views with indicated counts of each group. **c** Side views related to a 90° rotation of the cryoEM map of the HIV CA Di-hexamer at 3.14 Å resolution overlapped with the atomic models of subunits (colored). **d** Structural details of peripheral region of helix (aa: 48–81) in close-up views related to a 180° rotation. **e** The

design of mspSA affinity-capture of biotin-tagged DNA/RNA in complex with scPol II. **f** Representative 2D classes of scPol II EC generated from 1500 micrographs of mspSA affinity grid and holy carbon grid respectively applied same parameters for blob particle picking. 2D classes are separated into dominant- and tilted-views with indicated counts of each group. **g** Side views related to a 90° rotation of the cryoEM map of the scPol II EC at 2.98 Å resolution overlapped with the atomic models of subunits (colored). **h** Structural details of peripheral regions of DNA/RNA and helix-loop conjunction aa: (1191–1218) in close-up views.

was followed by further purification steps involving Hi-Trap Heparin and Mono Q anion exchange chromatography columns (Cytiva).

## Biotinylation of MCM and CA Di-hexamer

To enrich the MCM particles on SA grid, we first biotinylated MCM. The MCM protein was biotinylated at the lysine residues randomly via

reacting with NHS-PEG4-biotin or NHS-PEG12-biotin regents (Thermo Fisher) on ice for 2 hours followed by adding 50 mM Tris-HCl (pH 8.0) (final concentration) to quench the reaction. The free biotin molecules were removed by size exclusion chromatography (Superdex 200 lncrease 5/150 (GE)). The average number of biotin molecules covalently attached per MCM particle (MCM hexamer) is 1–2, a rate deemed

sufficient to minimize the labeling quantitated via the Biotin Quantitation Kit (Product No. 28005, Thermo Fisher Scientific)[38]. When comparing the structures of apo MCM (with biotinylation) and MCM-ATP-dsDNA complex (without biotinylation), we observed no significant local difference near the surface lysine sites, except for the pattern of WH which is likely due to DNA binding (as shown in Figs. 2g, 3f, S8c, and d).

Di-hexamer CA was biotinylated at the lysine residues randomly via reacting with NHS-PEG$_{12}$-biotin regents (Thermo Fisher) on ice for 2 hours followed by adding 50 mM Tris-HCl (pH 8.0) (final concentration) to quench the reaction. The free biotin molecules were removed by size exclusion chromatography (Superdex 200 Increase 5/150 (GE)). The average number of biotin molecules covalently attached per CA Di-hexamer is ~3, a rate deemed sufficient to minimize the labeling quantitated via the Biotin Quantitation Kit (Product No. 28005, Thermo Fisher Scientific)[38].

## Preparation of MCM-dsDNA complex
The 200 bp dsDNA substrates were generated from a pNIC-Bio2 plasmid and amplified by PCR via synthesized primers (ordered from Thermo Fisher Scientific). The primers used to make biotin labeled dsDNA for affinity grid were F_bio: 5′- biotin- atg aac atc aaa aag ttt gca aaa caa -3′, R_bio: 5′- biotin- tca gga act tga tat ttt tca ttt -3′; The primers used to make dsDNA as a control for the affinity grid are as follows: DNA: F: 5′- atg aac atc aaa aag ttt gca aaa caa -3′, R: 5′- tca gga act tga tat ttt tca ttt -3′. The sequence of the 200 bp dsDNA is: atg aac atc aaa aag ttt gca aaa caa gca aca gta tta acc ttt act acc gca ctg ctg gca gga ggc gca act caa gcg ttt gcg aaa gaa acg aac caa aag cca tat aag gaa aca tac ggc att tcc cat att aca cgc cat gat atg ctg g caa atc cct gaa cag caa aaa aat gaa aaa tat caa gtt cct ga. The PCR products were confirmed by agarose gel electrophoresis and these substrates with right molecular size were recovered via DNA purification kit (QIAGEN). The purified substrates were used to prepare MCM-ATP-dsDNA complex samples for cryoEM. The dsDNA and MCM were mixed with buffer containing 50 mM HEPES, PH 7.8, 150 mM NaCl, 1 mM MgCl$_2$, 1 mM ATP at 4°C and applied to the affinity grid.

## Preparation of ScPol II EC complex
PAGE-purified RNA and DNA oligonucleotides were obtained from Integrated DNA Technologies (IDT). The sequences used for preparing the elongation complex were as follows: biotinylated template strand DNA (tsDNA): 5′-/BiotinTEG/ TTT TTT GAT ATT TTT GGA TCC CGC TCT GCT CCT TCT CCC ATC CTC TCG ATG GCT ATG AGA TCA ACT AGG AAT TC-3′; biotinylated non-template strand DNA (ntsDNA) 5′-/BiotinTEG/ TTT TTA TGT ATT AAT GAA TTC CTA GTT GAT CTC ATA GCC ATC TCC TAC TTG GGA GAA GGA GCA GAG CGG GAT CC-3′; and 3′-deoxy RNA: 5′- AUC GAG AG/3′dG. To assemble the DNA/RNA scaffold, the RNA and tsDNA were first annealed by heating the mixture to 90 °C for 5 minutes and then allowing it to slowly cool to room temperature. The Pol II elongation complex (EC) was formed by mixing Pol II with the scaffold and incubating at 37 °C for 20 minutes. The ntsDNA was then added to the mixture and incubated at 37 °C for an additional 20 minutes. To mimic the ATP-incorporated state, ATP was added to the Pol II EC and incubated at room temperature for 10 minutes before grid freezing. The Pol II EC complexes were prepared in elongation buffer containing 20 mM Tris-HCl (pH 7.5), 40 mM KCl, 10 mM MgCl$_2$, and 5 mM dithiothreitol (DTT). The final concentrations of each component were 1 μM Pol II, 1.2 μM tsDNA, 1.5 μM ntsDNA, 1.2 μM RNA, and 5 mM ATP.

## 2-AP fluorescence assays
Double stranded DNA (66 bp, ordered from Sangon Biotech Co., Ltd (Shanghai, China)) contains the following sequence: cca aag atg aaa caa aag tat gaa aac aac taa gta cta aca atg aaa gaa tga aca aaa cat acg. On one side of the dsDNA, three positions were labeled with 2-AP. The first

position is at the 5$^{th}$ bp to the end which would bind around the WH motif of MCM. The second and the third position is at the 23$^{rd}$ bp and the 33$^{rd}$ bp. The experiments were carried out with F7000 (Hitachi High-Tech Corporation). The sample was excited at 315 nm so that there is little or no detectable contribution of fluorescence from aromatic amino acids of MCM at the emission peak of 2-AP at 368 nm. The buffer contained 20 mM HEPES PH 7.8, 150 mM NaCl, 1 mM MgCl$_2$. In addition, 0.6 mM AMPPNP or 0.6 mM ADP 0.6 mM ATP was added to the buffer. The observed fluorescence was corrected with buffer. The experiments were carried out with 400 nM DNA, 1600 nM MCM hexamer at 25 °C.

## CryoEM sample preparation using affinity grids
The steps to prepare the mspSA affinity enriched samples are as following:[1], Mix biotin labeled lipid (16:0 Biotinyl Cap PE, Avanti) and DOPC (18:1 (Δ9-Cis) PC (DOPC), Avanti) at the indicated ratios to the final concentration of 2 mg/ml; Prepare streptavidin (streptavidin from *Streptomyces avidinii*, Sigma) at the indicated concentrations in a buffer containing 50 mM HEPES, PH 7.2, 150 mM NaCl;[2], Take 28.4 μL streptavidin solution into the Teflon well (on ice) and then take 0.9 μL lipid mixture onto the surface of streptavidin solution; Enclose the Teflon block in a humility chamber and incubate for at least 2 hours; Lay the holy carbon grids (Quantifol 1.2/1/3 or 2/2, without glow discharge, onto the surface of Teflon well (carbon side facing downward, 2 mins) to pick up the lipid monolayer; Wash the grid with 3 drops (120 μL) of sample buffer to remove the free streptavidin (Supplementary Fig. 1);[3], Add 5-10 μL sample and incubate for about 15 mins, then wash with 1 drop (120 μL) of sample buffer to remove unattached particles;[4], Plunge-freeze the grids with Thermo Fisher (TF) Vitrobot IV (Time: 2.5 s, Force: -20, at 20 °C and 100% humidity). The buffer volume on the grids typically ranges from 2-4 μL. The blotting force should be optimized to achieve the best vitrification results. We selected a force of -20 in our Vitrobot. For quality control, the density of SA particles on the affinity grids would be examined (Supplementary Fig. 13). The mspSA method can tolerate a wide range of SA particle densities, as long as there are plenty of SA on the grids. Typically, four different MCM concentrations with duplicate grids were prepared. The grids were screened for ice thickness, monolayer intactness and MCM distribution, before data collection.

## CryoEM imaging
For the initial screening, we loaded cryo-grids into a Thermo Scientific™ Glacios™ transmission electron microscope, which was operated at 200 kV (Supplementary Table 2). The images were captured at a pixel size of 2.0 Å and total dose of 50 electrons per Å$^2$. Data were collected using EPU on a Titan Krios transmission electron microscope operated at 300 kV, equipped with Ametek-Gatan Bio-quantum-K3 detectors and energy filter or Falcon 4i detector & Selectris X energy filter. The respective imaging pixel sizes used in the data collection were 1.072 Å (Bio-quantum-K3) or 0.932 Å (Selectris X). The defocus ranged from −0.5 to −3.0 or −4.0 μm and the total dose was 40 or 39 electrons per Å$^2$ which is divided among 40 frames (Supplementary Table 3).

## Image processing and 3D reconstruction
For the MCM-apo sample, over 4000 raw movie micrographs were collected. Data processing was performed using CryoSPARC (v3.3.2)[52]. Following motion correction and patch CTF extraction of the movie frames, particles were selected by blob picking. Particles were then extracted using a box size of 384 pixels for multiple rounds of 2D classification. The particles from good 2D classes that showed MCM features were retained for further ab-initio reconstruction and heterogeneous refinement. Subsequently, 3D classes were performed followed by homogeneous 3D refinement. Finally, similar classes with high-resolution features were merged for the final homogeneous

refinement and non-uniform refinement. Local refinement was further carried out for each MCM subunit, achieving local resolutions ~3 Å for the four central MCM subunits. All locally refined parts were integrated into a single map using Chimera (v1.14)[80]. Data acquisition and processing are summarized in Supplementary Table 3. Detailed processing steps were outlined in Supplementary Fig. 4.

For the MCM-ATP-dsDNA sample, 14,584 raw movie micrographs were collected. Data processing was performed using CryoSPARC (v3.3.2)[52], similar to the MCM-apo sample, but with various rounds of global CTF refinement[81] and additional steps to eliminate classes lacking full dsDNA density in the map. Classification methods involved multiple rounds of 3D variability analysis and heterogeneous refinement. 3D classification was focused on dsDNA using a local mask. To accelerate the classification and conserve memory, particles were initially extracted at bin 3. After excluding corresponding particles containing no dsDNA density in the map, the remaining particles were extracted at bin 1 for further processing. Local refinement was conducted for each MCM subunit and dsDNA region to improve map resolution. All locally refined parts were integrated into a single map using Chimera (v1.14)[80]. In order to determine the configuration of double MCM hexamer, which is flexible, we masked off the density of CTD at the top MCM during alignmen. We were able to obtain a low-resolution map of stacked MCM hexamers. Reconstruction information is presented in Supplementary Table 3, and image processing and 3D reconstruction steps are depicted in Supplementary Fig. 5.

CA Di-hexamer and ScPol II EC were subject to a similar protocol for image processing via CryoSPARC (v3.3.2)[52]. Raw micrographs were imported, followed by motion correction and calculation of the contrast transfer function. The output micrographs were curated manually to remove those images with poor image quality (for example, astigmatic, moving). Then, 500 micrographs were used for automatic particle picking to generate initial 2D templates. After several rounds of 2D classification, those particles with the best 2D classes averages were selected for Topaz training[82] in CryoSPARC using the full dataset. After diverse 2D classification with different particle picking strategies, all good particles with decent 2D class averages were merged and duplicated particles were removed. The output particles were used for ab-initio reconstruction, followed by hetero-refinement. Good models with better overall structure features were selected for homo-refinement. Reconstruction information is presented in Supplementary Table 4, and image processing and 3D reconstruction steps are depicted in Supplementary Figs. 11 and 12.

### Atomic modeling, refinement, and validation

The model of a MCM subunit was predicted with AlphaFold2[83]. The model was then rigid body fitted into the best-resolved subunit (subunit of B or C) of each 3D cryoEM map with PHENIX (v1.14)[84] or Chimera (v1.14)[80]. The subunit models were first refined as rigid bodies with PHENIX, and subsequently adjusted manually in COOT (v 0.981)[85] guided by residues with bulky side chains. The model was then refined in real space by phenix.real_space_refine. Following this step, the best subunit models were used as an initial model to refine all other subunits of MCM with PHENIX and COOT. Finally, the models were validated using MolProbity[86]. Structural figures were prepared in ChimeraX (v1.5)[87] or Chimera (v1.14)[80].

The model of a CA subunit was predicted with 8TY6[58]. The model was then rigid body fitted into the one of the subunit with Chimera (v1.14)[80]. The model was first adjusted manually in COOT (v 0.981)[85] guided by residues with bulky side chains. The model was then refined in real space by phenix.real_space_refine. Following this step, all 12 subunits were fitted into the map and refined with PHENIX and COOT. Finally, the whole model was validated using MolProbity[86]. Structural figures were prepared in Chimera (v1.14)[80].

The model of all the subunits were predicted with 8TVY[59]. The model was then rigid body fitted into the map with Chimera (v1.14)[80].

The subunit models were first adjusted manually in COOT (v 0.981)[85] guided by residues with bulky side chains. The amino acids that are not well resolved are deleted. The model was then refined in real space by phenix.real_space_refine. Finally, the models were validated using MolProbity[86]. Structural figures were prepared in Chimera (v1.14)[80].

### Reporting summary

Further information on research design is available in the Nature Portfolio Reporting Summary linked to this article.

## Data availability

All data needed to evaluate the conclusions in the paper are present in the paper and/or the Supplementary information. Source Data are provided with this paper. The cryo-EM density maps and corresponding atomic models have been deposited in the EMDB and PDB, respectively. The accession codes are: for MCM-apo, EMD-38109 and PDB 8X7T (the composite map), EMD-38110 (the map before local refinement), EMD-38112, EMD-38113, EMD-38114, EMD-38115, EMD-38116, EMD-38117, EMD-38118 (maps after local refinement); for MCM-ATP-dsDNA: EMD-38111and PDB 8X7U (the composite map), EMD-38119 (the map before local refinement), EMD-38120, EMD-38121, EMD-38122, EMD-38123, EMD-38124, EMD-38125, EMD-38126, EMD-38127 (maps after local refinement); for CA Di-hexamer EMD-61286 and PDB 9JA0; for ScPol II EC: EMD-61287 and PDB 9JA1 Source data are provided with this paper.

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

## Acknowledgements

We thank Helen Duyvesteyn, Loic Carrique, Thomas Walter, Yanan Zhu, Tao Ni, Karen Davies, Lorna Malone, and Nathan Hardenbrook for their assistance in cryoEM data acquisition; J. Dong and Colin Freeman for computer system support; and Juan Shen for wetlab support; Keyu Lu for discussion on model validation. We thank Diamond Light source for access and support of the cryoEM facilities at the UK National Electron Bio-Imaging Centre (eBIC) (proposal NT29812). Further electron microscopy provision was provided through the OPIC electron microscopy facility, a UK Instruct-ERIC Centre, which was founded by a Wellcome JIF award (060208/Z/00/Z) and is supported by a Wellcome equipment grant (093305/Z/10/Z). Computation was performed at the Oxford Biomedical Research Computing (BMRC) facility, a joint development between the Wellcome Centre for Human Genetics (Wellcome Trust Core Award Grant Number 203141/Z/16/Z) and the Big Data Institute (BDI) supported by Health Data Research UK and the NIHR Oxford Biomedical Research Centre. This work was supported by the National Institutes of Health (U54 AI170791-7522, R21 CA280467, P.Z.), the UK Wellcome Trust Investigator Award 206422/Z/17/Z (P.Z.), the UK Biotechnology and Biological Sciences Research Council grant BB/S003339/1 (P.Z.) and ERC AdG grant 101021133 (P.Z.), National Key Research and Development Program of China [No. 2019YFA0709304 to M.L.]; National Natural Science Foundation of China [No. T2221001 to Y.L., No. 12104496, and 32371286 to J.M.].

## Author contributions

P.Z. conceived and supervised the project; G.Y. expressed and purified and labeled the MCM and CA proteins and DNA under supervision of R.J.C.G. and P.Z.; Q. L. expressed and purified the ScPol II EC under supervision of D. W.; J.M. developed the mspSA affinity grids, J.M. and G.Y. prepared the cryoEM samples; J.M., G.Y., and C.M. collected cryoEM data; J.M. and G.Y. performed the reconstructions; J.M., G.Y. and M.Y. built and validated the models; J.M., G.Y., and P.Z. carried out data analysis; P.Z., J.M., and G.Y. wrote the paper with contributions from all authors; Y.S., Y.L., and M.L. assisted during data analysis.

## Competing interests

The authors declare no competing interests.
