## [Transparent Peer Review file · Nature Communications]

Open Architecture of Archaea MCM and dsDNA Complexes Resolved Using Monodispersed Streptavidin Affinity CryoEM

Corresponding Author: Professor Peijun Zhang

Version 0:

Reviewer comments:

Reviewer #1

(Remarks to the Author)

The manuscript presents an improved affinity-based method for cryo-EM grid preparation, particularly suitable for enriching rare or dilute samples. The determined phase diagram will provide a general guide for others interested in the method. The improved method solves the particle orientation issue associated with the previous 2D lattice approach, but also introduces its own problem of not being able to readily remove the background contrast from the streptavidins. So, the usefulness of the method in its current form may be limited to large complexes that can tolerate slight contrast degradation. The authors have applied their method to study an archaeal homomeric MCM and found that it is a left-handed open spiral prior to or at the initial engagement with a dsDNA substrate. The eukaryotic Mcm2-7 heterohexamer when bound to Cdt1 has previously been reported to be a left-handed open spiral (ref 53). Therefore, this work lends support to the idea that mechanism of initial DNA melting inside the Mcm ring may be conserved between archaea and eukaryotes. The work is interesting, potentially useful. A revised version should address the following concerns.

Major concerns

1. Several major claims need to be toned down. For examples, the “new” or “novel” claim should be replaced with “improved” when describing their method. “High resolution” is not yet high enough resolution in most cases. Duplex melting is a strong claim that is not yet supported by a good enough EM density, hence speculative. The left-handed inactive open spiral configuration is not novel as both human and yeast Mcm2-7 exists in a similar form, although the extent of the gap is somewhat different.
2. DNA binding to the left-handed open spiral is expected to trigger gap closure and the conversion from left hand open to closed and largely flat or slightly right-handed configuration. I wonder if biotinylation of surface lysines has altered the enzyme activity such that it is no longer able to undergo the conversion. In vitro measurement to compare the ATP hydrolysis rate or DNA binding affinity or gel shift assay with or without the modification may tell.
3. The MCM-ATP-dsDNA complex, the authors should consider performing additional 3D classification with/without alignment focusing on the chamber DNA, to investigate whether the broad DNA density (7.8 Å local resolution) is due to melting or a result of flexible DNA positioning. The first three reasons listed in Discussion section (line290-296) do not convincingly support such speculation.
4. Dimerization of the MCM hexamer. This is a very interesting observation but needs better quality map to make any useful conclusion. This may be beyond the current study, but the authors should attempt additional methods e.g., crosslinking or GraFix, to improve the EM map.

Minor concerns

1. The word “novel” in title is inappropriate – rephrase with “improved”. Also, in abstract “new affinity-grid method” should be rephrased.
2. Line32, the authors should use more accurate description “high-resolution structures”, 3.3 and 3.6 Å cryoEM density maps are not yet high resolution. Also, in many other places through the text.
3. The figures should be cited in order.
4. Line 89, “We developed a new method” – We improved the streptavidin affinity grid preparation method?
5. Line 96 “We found both hexameric structures are left-handed spirals in open conformations, distinct from ALL previous reported conformations of archaea or eukaryote MCM helicases”: not true. Ref 53 described left open spiral for yeast Mcm2-7 (bound to Ctd1).
6. Line 98 “the dsDNA in the center channel of its MCM helicase complex appears in a rarely observed melted

conformation". This is an overstatement. The DNA density is not defined well enough to distinguish the two strands, hence it is unclear if the duplex is melted or not.

7. Line 175-177. yeast Mcm2-7 hexamer bound to Cdt1 was also found to be a left open ring (ref 53), and this should be included in discussion and comparison.

8. Line 179 "the central channel diameter of MCM-*apo* is ~50 Å". Many of the DNA binding elements in the chamber may be disordered (not resolved) in the absence of DNA, but they are still present and fill the chamber, so the actual diameter may be a lot smaller than 50 Å.

9. Line 236 "This region was clearly more flexible and less well resolved (7.81 Å resolution)", use more concise and accurate description – "this region is not well resolved"?

10. Line 314, "by passage through", passing through?

11. Line 324, "followed by adding 50 mM Tris-HCl (pH 8.0) to quench the reaction", please double check. The final concentration should be 50 mM for quenching?

12. Regarding the model validation statistics, the clashscore is 11.3 for the 3.6 Å resolution model and 16.3 for the 3.4 Å MCM *apo* model. This may be a bit high and may be further improved to around 10 at this resolution range.

Reviewer #2

(Remarks to the Author)

In this manuscript, the authors described an interesting method to assemble a streptavidin based affinity grid to enrich protein particles and minimize preferred orientation. They further elucidated this method with successful examples of mini-chromosome maintenance complex 3 and its complex with double strand DNA and nucleotide.

Overall I find the expansion of the existing streptavidin 2D crystal affinity grids clever and the structures resolved proved the usefulness of this method. But I have following questions that I would like the authors to clarify.

1. Is the *mspSA* affinity grids robust and reproducible? Although the method seems straightforward enough, past experience with streptavidin grids suggested that the process can be tricky. Can the grids be stored dry after assembly? And what characterization technique is used to carry out quality control? Answers to these questions will determine how many research labs will want to try this type of grids.
2. Line 74, the authors mentioned the high-resolution structure of MCM3 protein is still unknown. Can the authors briefly listed the previous efforts or why it has not been resolved?
3. I agree that in some scenarios, the streptavidin 2D crystal grids have rigid distances and will induce preferred orientations. But in the specific case of MCM3 protein, I suppose the rigidity could be mitigated by the random positions of biotin on the MCM3 proteins. Also, AWI is not an issue with the streptavidin 2D crystal grids. That being said, is there a reason why the *mspSA* grid is preferred here?
4. The authors mentioned that (line 133) that dsDNA has to be biotin tagged at both ends to prevent MCM from sliding off. Can the authors further explain on this? Are the dsDNA anchored onto the grids by both ends attached to two streptavidin like a bridge followed by the assembly of MCM? The illustration in Fig. 3A also showed one dsDNA only anchored to the grid by one end. Will sliding off happen? The control did prove that untagged dsDNA will not enrich MCM because there was likely no dsDNA on the *mspSA* grid. However, there was no experiment demonstrating the need to tag both ends of dsDNA. Besides, is it also viable to first attach MCM onto the grid followed by incubation with dsDNA? Please comment.
5. In Fig. 3b, in the image on the right, are the small particles in the background streptavidin? Why are they much denser than those in Fig. 2b? They are all images of samples without biotin on the *mspSA* grids.
6. Is it possible to calculate the density of SA particles on the affinity grids? Streptavidin particles are quite visible. That would be very beneficial as end users will have a good estimate that, in maximum, how many particles can be expected from their samples.

Reviewer #3

(Remarks to the Author)

In this manuscript, Ma et al. have developed a novel method for preparing streptavidin/biotin affinity-grids for cryo-EM structure determination. The authors have applied this method to study the TkMCM complexes and have obtained a series structures. Overall, the authors' new method shows promise in enriching challenging complexes on cryo grids for structural studies. However, while the structures of the TkMCM complexes obtained using this approach are informative, they do not offer significant new insights into the mechanisms that regulate MCM loading or DNA melting.

Despite the reviewer's enthusiasm for this work, there are several issues and concerns to be addressed in the present manuscript.

Major concerns:

1. The manuscript does not clearly explain the challenges of cryo sample preparation with Tk MCM complexes. It would be helpful for the authors to provide more information on why Tk MCM is particularly challenging to prepare for cryo sample and how it differs from other MCM complexes that has been studied using regular grids.
2. The manuscript does not provide clear information on the efficiency of surface lysine biotinylation and how many sites are modified per MCM complex. In addition, the potential impact of biotinylation on the overall shape and dynamics should be

addressed.

3. The authors' discovery that Tk MCM-*apo* forms a double hexameric ring with left-handed spiral is intriguing. The Tk MCM structures should be compared to other available MCM structures to generate unique insights into this particular MCM species and the mechanism underlying the formation of the soluble double hexamer. The authors should consider generating relevant mutants to disrupt DH formation and examine its impact on MCM loading and replication initiation.

4. The manuscript does not provide a clear explanation for how Tk MCM hexamers were loaded onto DNA. The authors should explain why MCM-*apo* can be loaded onto DNA into a double-hexamer form and should compare the structures between the *apo* form and the loaded form to generate insights into the MCM loading mechanism.

5. The authors should consider making use of the lipid monolayer binding sites for biotinylated DNA to study the MCM loading process directly on the cryo grid instead of mixing DNA and MCM in the test tube. This could involve imaging the loading process in a time-course manner on a grid bearing biotinylated DNA to generate insights into how Tk MCM is loaded onto DNA and how the loaded ones are fused into the DH form.

6. The manuscript collected 14,584 raw movie micrographs for the MCM-ATP-dsDNA sample, yet the final resolution was only resolved to 3.57 Å. The authors should consider testing this method with samples that are known to show preferred orientation preference using regular grids to determine whether the method is effective in improving structure determination.

7. The manuscript lacks other functional studies to support the structure, and the authors should consider including additional functional data to support their findings. The observation of melted/partially melted DNA is only a speculation due to the diffused density observed. Further functional studies should be conducted to confirm whether DNA is indeed melted by Tko MCM loading.

Version 1:

Reviewer comments:

Reviewer #1

(Remarks to the Author)

The authors have addressed my concerns in their nicely revised manuscript. This work will attract attention from both DNA replication field and cryo-EM method development.

Reviewer #2

(Remarks to the Author)

I'm satisfied with the authors' addressing of my concerns in the previous round and I'm happy to recommend publication with Nature Communications.

Some minor clarifications need to be made in the method section:

1. Manuscript line 499, I suppose the holey carbon grid is laid on to the lipid/SA film with carbon film facing downward. It should be clearly presented in the texts as well as the legends of supplementary Fig. 1.

2. Manuscript line 502 and 503, how much volume is used to wash the grid after sample incubation? And more importantly, how much volume of buffer is left on the grid for vitrification? Considering that a force of -20 was used already, there's not much room for optimization. The vitrification process may become a bottleneck for reproducibility.

Reviewer #3

(Remarks to the Author)

The authors have addressed the concerns raised by this reviewer. I recommend publication of this work.

Point-to-point responses to reviewers' comments

Reviewer #1

Major concerns

1. Several major claims need to be toned down. For examples, the “new” or “novel” claim should be replaced with “improved” when describing their method. “High resolution” is not yet high enough resolution in most cases. Duplex melting is a strong claim that is not yet supported by a good enough EM density, hence speculative. The left-handed inactive open spiral configuration is not novel as both human and yeast Mcm2-7 exists in a similar form, although the extent of the gap is somewhat different.

We appreciate the reviewer's comments. Accordingly, we have toned down the “terms” in the revised manuscript. We have also carried out additional experiments to support the structural implication, regarding duplex melting.

Furthermore, we have used this method to improve the preferred-orientation issue of two additional samples, namely HIV-1 capsid hexamer and Pol II elongation complex from yeast, and arrived at much better structures than previously achieved. These are included in the revised manuscript (See Figure 5).

2. DNA binding to the left-handed open spiral is expected to trigger gap closure and the conversion from left hand open to closed and largely flat or slightly right-handed configuration. I wonder if biotinylation of surface lysines has altered the enzyme activity such that it is no longer able to undergo the conversion. In vitro measurement to compare the ATP hydrolysis rate or DNA binding affinity or gel shift assay with or without the modification may tell.

We thank the reviewer for this comment. As shown in Figure 3a, in MCM-dsDNA complexes, surface lysine of MCM was not biotinylated, instead, the biotin is attached to the end of dsDNA. The left-handed open conformation appears a property of both MCM-apo and MCM-dsDNA complexes with or without biotinylation of surface lysine.

3. The MCM-ATP-dsDNA complex, the authors should consider performing additional 3D classification with/without alignment focusing on the chamber DNA, to investigate whether the broad DNA density (7.8 Å local resolution) is due to melting or a result of flexible DNA positioning. The first three reasons listed in Discussion section (line290-296) do not convincingly support such speculation.

We thank the reviewer for this comment. We have now performed additional 3D classification without alignment focusing on the chamber DNA, in addition to the previous 3D classification with alignment. Unfortunately, both methods showed a broad DNA density. The additional classification are included in the supplementary Figure 5.

To investigate the dsDNA conformation in the MCM binding pocket, we further carried out fluorescence assays. We used 2-aminopurine (2-AP):T base pairing at different locations of dsDNA and measured the intensity of 2-AP in the presence of different nucleotides. The results support opening of dsDNA, and are included in the revised manuscript (page 9) and new figure panels, Figure 4d-f.

4. Dimerization of the MCM hexamer. This is a very interesting observation but needs better quality map to make any useful conclusion. This may be beyond the current study, but the authors should attempt additional methods e.g., crosslinking or GraFix, to improve the EM map.

We appreciate the reviewer's suggestion. We will follow up on the dimerization of the MCM hexamer in our future studies.

Minor concerns

1. The word "novel" in title is inappropriate – rephrase with "improved". Also, in abstract "new affinity-grid method" should be rephrased.

We have rephrased the above words.

2. Line32, the authors should use more accurate description "high-resolution structures", 3.3 and 3.6 Å cryoEM density maps are not yet high resolution. Also, in many other places through the text.

We have rephrased the words to make it more accurate.

3. The figures should be cited in order.

We thank the reviewer for the comment and have cited the figures in order.

4. Line 89, "We developed a new method" – We improved the streptavidin affinity grid preparation method?

We appreciate the reviewer's suggestion and have rephrased the above words.

5. Line 96 "We found both hexameric structures are left-handed spirals in open conformations, distinct from ALL previous reported conformations of archaea or eukaryote MCM helicases": not true. Ref 53 described left open spiral for yeast Mcm2-7 (bound to Ctd1).

We thank the reviewer for the comment and have revised the above sentence as "We found both hexameric structures are left-handed spirals in a rare open conformation." The previously reported open conformation in yeast Mcm2-7 (bound to Ctd1) (ref 53

in the last version, ref 70 in the current version) has a much smaller opening gate, 6 Å, compared to 15-20 Å in this study.

6. Line 98 “the dsDNA in the center channel of its MCM helicase complex appears in a rarely observed melted conformation”. This is an overstatement. The DNA density is not defined well enough to distinguish the two strands, hence it is unclear if the duplex is melted or not.

We appreciate the reviewer’s comment. Indeed, DNA density is not defined well enough to indicate a melted conformation.

To further explore the conformation of dsDNA in the MCM binding pocket, we used 2-AP:T to measure the change of fluorescence signal upon MCM loading at different base pair positions and in the presence of different nucleotide analogues (See response to Major Point 3). We have included new results in the revised manuscript (Figure 4d-f).

7. Line 175-177. yeast Mcm2-7 hexamer bound to Cdt1 was also found to be a left open ring (ref 53), and this should be included in discussion and comparison.

We appreciate the reviewer’s comment. We have included this work in Table 1, discussion and comparison in the main text (page 12).

8. Line 179 “the central channel diameter of MCM-apo is ~50 Å”. Many of the DNA binding elements in the chamber may be disordered (not resolved) in the absence of DNA, but they are still present and fill the chamber, so the actual diameter may be a lot smaller than 50 Å.

We thank the reviewer for the comment. WH motifs that locate at the end subunits were not resolved well, but residues in the center channel are mostly resolved for modeling the structure. Nonetheless, some of the sidechains maybe flexible without DNA substrate. Thus, we measured the distance between the backbone of two amino acids at the same Z position at CTD for the size of channel consistently in all cases. Now we have clarified the distance measurement into the table, main text (page 7 and 8), and figure legends.

9. Line236 “This region was clearly more flexible and less well resolved (7.81 Å resolution)”, use more concise and accurate description – “this region is not well resolved”?

We have revised the above sentence to make it more accurate.

10. Line 314, “by passage through”, passing through?

We have corrected it.

11. Line 324, “followed by adding 50 mM Tris-HCl (pH 8.0) to quench the reaction”, please double check. The final concentration should be 50 mM for quenching?

Yes, 50 mM is the final concentration. Thanks for pointing this out. We have revised it to make it clear.

12. Regarding the model validation statistics, the clash score is 11.3 for the 3.6 Å resolution model and 16.3 for the 3.4 Å MCM apo model. This may be a bit high and may be further improved to around 10 at this resolution range.

Many thanks for the suggestion. We have refined the model and improved the clash score better than 10 (MCM-apo is ~4.89, MCM-ATP-dsDNA is ~9.75).

Reviewer #2:

1. Is the mspSA affinity grids robust and reproducible? Although the method seems straightforward enough, past experience with streptavidin grids suggested that the process can be tricky. Can the grids be stored dry after assembly? And what characterization technique is used to carry out quality control? Answers to these questions will determine how many research labs will want to try this type of grids.

We thank the reviewer for the comment. The method is straight forward, robust and reproducible. In our experience, mspSA is easier to make than the 2D crystalline SA. But lipid monolayer needs to be handled with care.

The current version of the grids can't be stored dry after assembly. However, the grid may be stored dry upon coating the back side of the grid with a graphene or thin carbon layer with trehalose-embedding of lipid monolayer (DOI:10.1016/j.jsb.2016.06.009).

For quality control (as the reviewer suggested in point 6), one would check the density of SA particles on the affinity grids (Supplementary Figure 13). The mspSA method can tolerate a wide range of SA particle densities. We typically make grids with four different MCM concentrations, each repeated with 2 grids. We screened the grids for ice thickness, monolayer intactness and MCM distribution, before data collection. We have expanded the methods section to include these (page 16).

2. Line 74, the authors mentioned the high-resolution structure of MCM3 protein is still unknown. Can the authors briefly listed the previous efforts or why it has not been resolved?

There have been many efforts to determine the structure of substrate-bound Archaea MCM via crystallography or cryoEM but with little success. Recently, Meagher *et al.*

reported cryoEM structures of truncated and modified *Saccharolobus solfataricus* MCM (SsoMCM) in complex with ssDNA to improve sample stability (DOI: 10.1038/s41467-019-11074-3; DOI: 10.3390/ijms232314678). However, the challenged structure of MCM in complex with initial substrate dsDNA is still unknown.

The main reason making this complex structure difficult to determine is that the MCM can slide on the dsDNA in the presence of ATP (DOI: 10.1016/j.cell.2009.10.015; DOI: 10.1073/pnas.1712537114). Previous work on other MCM complexes have used various strategies to deal with the issue of sliding off dsDNA substrate, such as using ringed dsDNA (DOI: 10.1016/j.cell.2009.10.015), end-blocked dsDNA (DOI: 10.1038/s41594-021-00698-z), or optimized purification method (DOI: 10.1073/pnas.1712537114). Our initial experiment using native gels also demonstrated dissociation of the MCM-dsDNA complex.

To solve the above issues, we devised SA affinity grids to not only block both ends of the dsDNA preventing MCM sliding off, but also to selectively enrich MCM-loaded-dsDNA complexes for the cryoEM sample.

We have added above content in the revised manuscript (page 4).

3. I agree that in some scenarios, the streptavidin 2D crystal grids have rigid distances and will induce preferred orientations. But in the specific case of MCM3 protein, I suppose the rigidity could be mitigated by the random positions of biotin on the MCM3 proteins. Also, AWI is not an issue with the streptavidin 2D crystal grids. That being said, is there a reason why the mspSA grid is preferred here?

There are several reasons why the mspSA grid is preferred here. 1) The method is more robust and simpler, easy to adapt; 2) mspSA method can tolerate a wider range of conditions, including SA concentration, buffer condition, and better coverage on the grid; 3) CryoEM SPA data processing is simpler, as the SA density is usually averaged out, while for the SA 2D lattice, additional 2D Fourier filtering of SA lattice diffraction spots is needed to remove the SA signal; 4) The reconstruction steps (such as Bayesian polishing in Relion) with the raw movie frames could be directly used in mspSA method. We have added the above description in discussion (page 11).

4. The authors mentioned that (line 133) that dsDNA has to be biotin tagged at both ends to prevent MCM from sliding off. Can the authors further explain on this? Are the dsDNA anchored onto the grids by both ends attached to two streptavidin like a bridge followed by the assembly of MCM? The illustration in Fig. 3A also showed one dsDNA only anchored to the grid by one end. Will sliding off happen? The control did prove that untagged dsDNA will not enrich MCM because there was likely no dsDNA on the mspSA grid. However, there was no experiment demonstrating the need to tag both ends of dsDNA. Besides, is it also viable to first attach MCM onto the grid followed by

incubation with dsDNA? Please comment.

We appreciate the reviewer's comment. In the presence of ATP, the MCM could slide on and run off the dsDNA template (DOI: 10.1016/j.cell.2009.10.015; DOI: 10.1073/pnas.1712537114). Other studies have used very long dsDNA, end-blocked dsDNA, and circular dsDNA to circumvent the problem (DOI: 10.1016/j.cell.2009.10.015, DOI: 10.1038/s41594-021-00698-z, DOI: 10.1073/pnas.1712537114). We designed the experiments to tag both ends of dsDNA on the following rationale: 1) to selectively enrich the dsDNA-bound MCM full complex on the grid; 2) to prevent MCM running off dsDNA when ATP is present. Although there was no experiment demonstrating the need to tag both ends of dsDNA, we thought tagging both ends would help to prevent MCM sliding off dsDNA substrate.

We mixed MCM, ATP and dsDNA and added the mixture to the mspSA monolayer. There might still be a fraction of complexes with only one-end attached, as illustrated in the original Figure 3A. If sliding off happens, MCM will likely not be retained on the grid. This way, the full MCM-ATP-dsDNA complex is favored on the mspSA affinity grid.

It might be viable to first attach MCM onto the grid followed by incubation with dsDNA, but not with ATP. In the present of ATP, MCM will slide off the dsDNA substrate, based on the previous reports and our experiments. Additionally, there will be a mixture of MCM with and without dsDNA, making data processing more challenging. Using biotin-tagged dsDNA tethered to the mspSA grids, the MCM-loaded particles will be selectively enriched. We have revised the text accordingly to include these (page 4).

5. In Fig. 3b, in the image on the right, are the small particles in the background streptavidin? Why are they much denser than those in Fig. 2b? They are all images of samples without biotin on the mspSA grids.

Yes, the small particles in the background are streptavidin. It's denser in Fig 3b than that in Figure 2b because of a higher SA concentration was used (0.04 mg/ml versus 0.02 mg/ml). Now we have specified the concentration of SA in the revised figure legend.

6. Is it possible to calculate the density of SA particles on the affinity grids? Streptavidin particles are quite visible. That would be very beneficial as end users will have a good estimate that, in maximum, how many particles can be expected from their samples.

We appreciate the reviewer's suggestion. Indeed, SA particles are visible and can be counted. We have added SA particle distribution as a quality control in the method of the revised manuscript (Page 16) and as a supplementary figure S13.

Reviewer #3:

Major concerns:

1. The manuscript does not clearly explain the challenges of cryo sample preparation with Tko MCM complexes. It would be helpful for the authors to provide more information on why Tko MCM is particularly challenging to prepare for cryo sample and how it differs from other MCM complexes that has been studied using regular grids.

The MCM can slide on the dsDNA in the presence of ATP (DOI: 10.1016/j.cell.2009.10.015; DOI: 10.1073/pnas.1712537114). Previous work on other MCM complexes had to design various strategies to deal with the issue of sliding off dsDNA substrate, such as using ringed dsDNA (DOI: 10.1016/j.cell.2009.10.015) , end-blocked dsDNA (DOI: 10.1038/s41594-021-00698-z), or optimized purification method (DOI: 10.1073/pnas.1712537114). Our initial experiment using native gels also demonstrated the issue with dissociation of the MCM-dsDNA complex.

To solve this problem, we devised SA affinity grids to not only block both ends of the dsDNA preventing MCM sliding off, but also to selectively enrich MCM-loaded-dsDNA complexes for the cryoEM sample.

We have added the above rationale in the Introduction (page 4).

2. The manuscript does not provide clear information on the efficiency of surface lysine biotinylation and how many sites are modified per MCM complex. In addition, the potential impact of biotinylation on the overall shape and dynamics should be addressed.

With the Biotin Quantitation Kit, the average number of biotin molecules covalently attached per MCM particle (MCM hexamer) is 1-2, thus minimizing potential effect of surface modification. We have added this in the method section (page 14), along with a new supplementary figure panel (Supplementary Figure 3b).

By comparing the structures of apo MCM (with biotinylation) and MCM-ATP-dsDNA complex (without biotinylation), we observed no significant local difference near the surface lysine sites, except for the pattern of WH which is likely due to DNA binding (as shown in Figure 2g, 3f, S8c, and d).

3. The authors' discovery that Tk MCM-apo forms a double hexameric ring with left-handed spiral is intriguing. The Tk MCM structures should be compared to other available MCM structures to generate unique insights into this particular MCM species and the mechanism underlying the formation of the soluble double hexamer. The authors should consider generating relevant mutants to disrupt DH formation and examine its impact on MCM loading and replication initiation.

We compared Tko MCM with other MCM structure as shown in Figure 2g, h and 3f, g. They showed the conformation differences in MCM monomers, resulting in the differences of the whole hexamer arrangement, including the handedness. We have added a new Supplementary Figure 9 to illustrate the differences and revised the text accordingly (Page 9).

For the formation of double hexamers, it was reported that the Zinc binding motif is important (DOI:10.1038/nsb893). Previous function studies have shown the mutation of the Zinc binding motif disrupted double hexamer formation in MthMCM ((DOI: <https://doi.org/10.1074/jbc.M509773200>) and resulted in lethality or temperature sensitivity in yeast (DOI: 10.1128/MCB.17.10.5867; DOI: 10.1101/gad.5.6.944).

We appreciate the reviewers' suggestion on generating relevant mutants to disrupt DH formation and examine its impact on MCM loading and replication initiation. We plan to explore this in future studies.

4. The manuscript does not provide a clear explanation for how Tk MCM hexamers were loaded onto DNA. The authors should explain why MCM-apo can be loaded onto DNA into a double-hexamer form and should compare the structures between the apo form and the loaded form to generate insights into the MCM loading mechanism.

We compared the structures between Tko MCM apo and dsDNA-bound MCM. Unfortunately, the dsDNA substrate is poorly resolved and its interaction with MCM is not clear, thus not allowing us to provide a clear view of interactions between Tko MCM hexamers and dsDNA. Future structural studies are needed to understand MCM loading.

5. The authors should consider making use of the lipid monolayer binding sites for biotinylated DNA to study the MCM loading process directly on the cryo grid instead of mixing DNA and MCM in the test tube. This could involve imaging the loading process in a time-course manner on a grid bearing biotinylated DNA to generate insights into how Tk MCM is loaded onto DNA and how the loaded ones are fused into the DH form.

We appreciate the reviewer's comments. Indeed, we are planning such study for future work, especially in the light of a recent exciting time-resolved cryoEM study by Thomas C. R. Miller *et.al.* (DOI: 10.1038/s41586-019-1768-0). It is very interesting, but we think this is beyond the scope of current study.

6. The manuscript collected 14,584 raw movie micrographs for the MCM-ATP-dsDNA sample, yet the final resolution was only resolved to 3.57 Å. The authors should consider testing this method with samples that are known to show preferred orientation

preference using regular grids to determine whether the method is effective in improving structure determination.

The MCM-ATP-dsDNA complexes are likely flexible and dynamic (ATP hydrolysis cycle), which limits the current resolution despite a large number of movies were included.

We appreciate the reviewer's suggestion, and have included two additional samples, crosslinked HIV-1 CA hexamer and RNA Pol II elongation complex, both had been challenged by the preferred-orientation. Previously a tilted dataset was needed for HIV-1 CA hexamer. The RNA Pol II elongation complex had limited resolution due to severe preferred orientation problem. Using mspSA affinity grids, we now determined the CA hexamer SPA structure to 3.14 Å resolution and Pol II EC to 2.98 Å resolution. New results are included in the revised manuscript (page 10 and 11) and with an additional Figure 5 (also shown below Figure R1).

Figure R1 | CryoEM structures of HIV-1 CA Di-hexamer and scPoll EC complex using mspSA grids. (a) The design of mspSA affinity-capture of biotinylated HIV CA Di-hexamer. (b) Representative 2D classes of HIV-1 CA Di-hexamer. (c) The cryoEM map of the HIV CA Di-hexamer at 3.14 Å resolution overlapped with the atomic models of subunits (colored). (d) Structural details of peripheral region of helix (48-81) in close-up views. (e) The design of mspSA affinity-capture of biotin-tagged DNA/RNA in complex with scPoll. (f) Representative 2D classes of scPoll EC. (g) The cryoEM map of the scPoll EC at 2.98 Å resolution overlapped with the atomic models of subunits (colored). (h) Structural details of peripheral regions of DNA/RNA and helix-loop conjunction (1191-1218) in close-up views.

7. The manuscript lacks other functional studies to support the structure, and the authors should consider including additional functional data to support their findings. The observation of melted/partially melted DNA is only a speculation due to the diffused density observed. Further functional studies should be conducted to confirm whether DNA is indeed melted by Tko MCM loading.

We appreciate the reviewer's comments. We have conducted further studies to test the hypothesis of melted/partially melted DNA. Inspired by previous studies (DOI: 10.1073/pnas.1212929109; DOI: 10.7554/eLife.06562), we designed a similar experiment to measure the local melting of the dsDNA. On the template of 66 bp dsDNA, three positions were labeled separately with 2-AP. The first position is at the

5th bp, close to the end which would bind around the WH motif of MCM. The second and third positions are at the 23rd and 33rd bp in the channel. Fluorescence data suggest potential melting of dsDNA at the 23rd and 33rd bp positions. The details are now described in the main text (page 9) with additional figure panels (Figure 4d-f) and Supplementary Figure 10.

Point-to-point responses to reviewers' comments:

Reviewer #2 (Remarks to the Author)

I'm satisfied with the authors' addressing of my concerns in the previous round and I'm happy to recommend publication with Nature Communications.

Some minor clarifications need to be made in the method section:

1. Manuscript line 499, I suppose the holey carbon grid is laid on to the lipid/SA film with carbon film facing downward. It should be clearly presented in the texts as well as the legends of supplementary Fig. 1.

Yes, the reviewer is correct. The carbon side is laid onto the monolayer. Now we have added the above information to the text as well as the legend of supplementary Fig. 1.

2. Manuscript line 502 and 503, how much volume is used to wash the grid after sample incubation? And more importantly, how much volume of buffer is left on the grid for vitrification? Considering that a force of -20 was used already, there's not much room for optimization. The vitrification process may become a bottleneck for reproducibility.

We appreciate the reviewer's comment. The volume used to wash the grid after sample incubation is 120 ul. We have added the above information in the text.

The buffer volume left on the grid typically ranges from 2-4 μ l. The blotting force should be optimized to achieve the best vitrification results. We selected a force of -20 in our Vitrobot. This information has now been incorporated into the text.